# Design and Analysis of High-Capacity MIMO System in Line-of-Sight Communication

**DOI:** 10.3390/s22103669

**Published:** 2022-05-11

**Authors:** Kuangming Jiang, Xiaoyong Wang, Yanliang Jin, Asad Saleem, Guoxin Zheng

**Affiliations:** 1Key Laboratory of Specialty Fiber Optics and Optical Access Networks, Shanghai University, Shanghai 200444, China; jiangkuangming@163.com (K.J.); wuhaide@shu.edu.cn (Y.J.); 2CASCO Signal Ltd., Shanghai 200071, China; wangxiaoyong@casco.com.cn; 3Key Laboratory of Antennas and Propagation, Shenzhen University, Shenzhen 518060, China; asadalvi64@yahoo.com

**Keywords:** LoS, MIMO, channel capacity, antenna configuration

## Abstract

The phase of the channel matrix elements has a significant impact on channel capacity in a mobile multiple-input multiple-output (MIMO) communication system, notably in line-of-sight (LoS) communication. In this paper, the general expression for the phase of the channel matrix at maximum channel capacity is determined. Moreover, the optimal antenna configuration of the 2 × 2 and 3 × 3 transceiver antenna array is realized for LoS communication, providing methods for n×n optimal antenna placement, which can be used in short-range LoS communication and non-scattering environment communication, such as coupling train communication and inter-satellite communication. Simulation results show that the 2 × 2 rectangular antenna array is more suitable for the communication of coupling trains, while the 3 × 3 circular arc antenna array is more suitable for virtual coupling trains according to antenna configurations. Moreover, the 2 × 2 antenna rectangular configuration proposed in this paper has reached the optimal channel in inter-satellite communication, which lays a foundation for the deployment of communication systems.

## 1. Introduction

In recent years, the fifth generation (5G) mobile communication system has evolved as a valuable tool to meet the rapid development of MIMO communication. The 5G wireless communication is used in a wide range of emerging applications to meet highly stringent latency, dependability, mobility, and security requirements. However, with the improvement of application demands, more stringent latency and reliability are required in fields such as unmanned driving and remote surgery [1,2,3], so it is necessary to develop towards the sixth generation (6G) mobile communication. In almost all application scenarios, low cost, low power consumption, and high stability are common requirements, which undoubtedly become the challenge of mobile communication.

MIMO technology is one of the key technologies used for mobile communication, which can greatly improve the capacity and spectrum utilization of communication systems without increasing the bandwidth. MIMO technology is widely used in wireless communication, such as Internet of Things (IoT) and wireless sensor networks (WSNs). Antenna transmission, both in IoT and WSNs, is an important part of a wireless communication system, and the antenna structure of the MIMO system directly affects the transmission performance. In order to optimize the channel transmission performance, this paper derives the relationship between the channel matrix phase and channel capacity, and designs the optimal transceiver antenna configuration.

### 1.1. Motivation and Contributions

The existing research primarily focuses on the theoretical study of small-scale LoS MIMO systems; however, there are a few studies that can be employed in a variety of practical LoS MIMO applications, such as coupling train communication and satellite communication. In these application scenarios, the antenna configuration often results in poor channel characteristics. Therefore, this paper derives the precise expression between channel phase and capacity of the n-order MIMO system in LoS communication. Since there exists a small amplitude difference in channel matrix elements, the phase of the channel matrix is regarded as the most critical component for the channel capacity and the optimal antenna configurations. The theoretical premise is a channel without scattering and multipath, which is mainly used in short-range LoS communication and non-scattering environment communication, such as coupling train communication and inter-satellite communication. Moreover, the optimal placement and parameters of 2 × 2 and 3 × 3 transceiver antennas are provided based on the maximum channel capacity in coupling train communication and inter-satellite communication, which lays a basic foundation for the MIMO layout.

### 1.2. Related Literature

There are many emerging technologies in the field of new generation mobile communication. Terahertz (THz) communication has emerged as a possible alternative to accommodate a future 6G wireless network massive data load and increase network capacity. Air radio access networks (ARANs) support the global access infrastructure for seamless mobile communication systems [4]. When compared to the sub-6 GHz massive MIMO system, the aerospace terahertz MIMO channel has a unique delay-beam–doppler squint effect, and transmissions are mostly along the LoS direction [5]. Satellite communication frequency has been addressed mostly on the Ka frequency range (26 GHz–40 GHz), where more efforts are needed in the U and V higher frequency bands (40 GHz–75 GHz) [6]. The authors of [7] focused on satellite massive MIMO transmission and stated that a massive MIMO transmission scheme based on frequency reuse can boost the data rate of satellite communication systems. The authors in [8] first provided an overview of green transportation in automotive industry 5.0 and showed significance of backscatter communication and non-orthogonal multiple access (NOMA) for green transportation applications. A novel alternating optimization framework is proposed in [9] to enhance the energy efficiency of intelligent reflective surface (IRS)-assisted non-orthogonal multiple access beamforming (NOMA-BF) systems for 6G wireless communication networks. The authors in [10] provided an overview of an IRS-enhanced physical layer security (PLS) for unmanned aerial vehicle (UAV) communication. The IRS-assisted PLS is presented in UAV communications to improve the secrecy rate.

Many studies on MIMO technology have been conducted in recent years, including capacity analysis, channel estimation, and channel modeling [11]. An effective solution, based on different antenna parameters to boost the massive MIMO channel capacity, is provided in [12]. Under low signal-to-noise ratio (SNR) situations, the authors in [13] derived a simple analytical expression for the MIMO channel capacity with Rayleigh fading. In [14], it the authors proved that MIMO channel capacity drops with the increase of K-factor, and increases with the increase of delay spread. The statistical comparison of curved and straight tunnels on channel performance is analyzed in [15]. Changes in K-factor over time, frequency, and location are investigated in [16]. The relationship between Rician channel and K-factor, as well as the effects of mobility and the carrier on K-factor is studied in [17]. Statistical parameters of the channel, such as root mean square delay spread (RMS-DS), K factor, and shadowing are studied in [18,19]. The efficient method to approximate the MIMO channel model by sparse matrices with an equivalent Markov process is introduced and thoroughly analyzed in [20]. A shared aperture two-element MIMO antenna design for 5G standards is presented in [21], which uses the same radiating structure to cover both the sub-6 GHz and millimeter-wave bands. In [22], the analytic upper bound and lower bound of the ergodic capacity are presented for distributed MIMO systems operating on Rayleigh/lognormal fading channels. The influence of positive definiteness of the matrix on channel correlation under realistic propagation conditions in MIMO wireless channels is studied in [23,24]. The derivation and analysis of spatial correlations based on stochastic channel models for MIMO systems using polarized and isolated antenna arrays are studied in [25,26]. The authors in [27] eliminated all numerical integrations required to compute far-field envelope cross-correlation in a MIMO system, deriving accurate and efficient analytical expressions for the frequency-domain cross-correlation Green’s function. The authors in [28,29] proposed research of a linearly polarized compact multi-band MIMO antenna system for small mobile terminals, and conducted an in-depth analysis of its antenna circuit. The effect of the antenna position on MIMO communication in LoS and NLoS system is studied in [30] using the method of moments and the finite-difference time-domain. The authors in [31] have derived analytical expressions for the achievable rate of asymptotic ergodicities of the system under a zero-forcing (ZF) detector. The capacity of a 2 × 2 short-range LoS MIMO channel in three-dimensional space is studied and the spherical wave model (SWM) is used to model the corresponding channel with arbitrary antenna directions in three-dimensional space in [32,33,34].

There are some literature studies related to short-range communication. In [35], the transmit design for short-range MIMO channels with the SWM is studied and the analytical expression of phase differences is derived, which makes the channel matrix optimally well-conditioned. A LoS MIMO channel model is proposed in [36], which considers an additional phase in the transmission path. The channel performance can be improved by adding a dielectric material. A constraint for antenna arrangement is designed as a function of frequency and distance for the LoS MIMO communication in [37]. The authors in [38] performed a novel analysis of the sensitivity of the optimal design parameters and derived analytical expressions for the eigenvalues of the pure LoS channel matrix. Two channel phase measurement methods of microwave LoS MIMO systems are investigated in [39]. In [40], the SWM enables the performance of the short-range LoS MIMO system to be significantly improved by properly adjusting the array geometries. According to the authors of [41], when the distance between the transmitting and receiving array antennas is short enough, the array antenna with the optimally narrow element spacing can give low spatial correlation.

### 1.3. Article Structure

The paper is structured as follows. Section 2 derives theoretical expressions for the channel phase and capacity of n-order elements. Section 3 analyzes the optimal placement of 2 × 2 and larger transceiver antennas based on the maximum capacity theory. Section 4 provides a comparative analysis of the performance of different antenna configurations. Section 5 provides physical insight and applications of the MIMO system in coupling train communication and inter-satellite communication. Section 6 draws the conclusion of this paper.

## 2. Theory of Channel Capacity

The input–output relationship of communication systems in a narrow-band Gaussian channel can be expressed based on the following expression:(1)y=H·x+n
where H is the channel transmission matrix with the size of M×N. y is a vector of M receiving antennas, x is a vector of N transmitting antennas, n is Gaussian noise.

The channel matrix is normalized by Frobenius norm to eliminate the influence of large-scale fading. It can be defined as:(2)$$H_{nor}=H\cdot \sqrt{\frac{M\times N}{{\left|\left|H\right|\right|}_{F}^{2}}}$$
where ${\left|\left|H\right|\right|}_{F}$ is the Frobenius norm of H. The Frobenius norm is defined in the complex number domain, as follows:(3)$${\left|\left|H\right|\right|}_{F}=\sqrt{\overset{M}{\underset{i=1}{\sum }}\overset{N}{\underset{j=1}{\sum }}{\left|h_{ij}\right|}^{2}}$$
where $h_{ij}$ is the (i,j) th element in channel matrix H.

In MIMO systems, if the transmitter has no channel state information and each antenna transmits the same power in the equivalent MIMO channel model [42], the total channel capacity can be defined as:(4)$$C=log_{2}\det\left(I_{M}+\frac{\rho }{N}H_{nor}H_{nor}^{*}\right)$$
where det is the determinant. $H_{nor}^{*}$ is the conjugate transpose of $H_{nor}$. ρ is SNR, which is defined as the ratio of the sum of the received power and the noise power.

The determinant calculation approach can be used to explore the phase characteristics of the channel matrix H at maximum channel capacity of the n-order MIMO system. When the number of receiving and transmitting antennas is the same, the n×n MIMO channel matrix can be defined as:(5)$$H={\left(\begin{array}{cc} \begin{array}{cc} A_{11}e^{j\varphi_{11}} & A_{12}e^{j\varphi_{12}}\\ A_{21}e^{j\varphi_{21}} & A_{22}e^{j\varphi_{22}} \end{array} & \begin{array}{cc} \cdots & A_{1n}e^{j\varphi_{1n}}\\ \cdots & A_{2n}e^{j\varphi_{2n}} \end{array}\\ \begin{array}{cc} \vdots & \vdots \\ A_{n1}e^{j\varphi_{n1}} & A_{n2}e^{j\varphi_{n2}} \end{array} & \begin{array}{cc} \ddots & \vdots \\ \cdots & A_{nn}e^{j\varphi_{nn}} \end{array} \end{array}\right)}_{n\times n}$$
where $\varphi_{ij}$ is the phase of the channel matrix element and the size of H is n×n. Generally, the distance of the transceiver antennas is quite larger than the spacing of the antenna array elements, resulting in identical amplitudes for the elements in the channel matrix. Therefore, the received power can be considered the same, which means $A_{ij}=1\left(1\leq i\leq n,1\leq j\leq n\right)$, and the phase becomes the main factor that has critical influence on the channel capacity. Therefore, $\det\left(I_{n}+\frac{\rho }{n}HH^{*}\right)$ can be calculated as:(6)$$\det(I_{n}+\frac{\rho }{n}HH^{*})=|\begin{array}{cccc} \rho +1 & \frac{\rho }{n}\sum_{k=1}^{n}e^{j\left(\varphi_{1k}-\varphi_{2k}\right)} & \ldots & \frac{\rho }{n}\sum_{k=1}^{n}e^{j\left(\varphi_{1k}-\varphi_{nk}\right)}\\ \frac{\rho }{n}\sum_{k=1}^{n}e^{j\left(\varphi_{2k}-\varphi_{1k}\right)} & \rho +1 & \ldots & \frac{\rho }{n}\sum_{k=1}^{n}e^{j\left(\varphi_{2k}-\varphi_{nk}\right)}\\ \vdots & \vdots & \ddots & \vdots \\ \frac{\rho }{n}\sum_{k=1}^{n}e^{j\left(\varphi_{nk}-\varphi_{1k}\right)} & \frac{\rho }{n}\sum_{k=1}^{n}e^{j\left(\varphi_{nk}-\varphi_{2k}\right)} & \ldots & \rho +1 \end{array}|$$

It can be seen from Equation (6) that the determinant follows the Hermitian symmetric property. When the elements of the non-main diagonal are all equal to zero, the determinant reaches the maximum value of ${\left(\rho +1\right)}^{n}$.

Further analysis shows that each non-main diagonal element in the determinant can be regarded as the sum of n vectors. Since the modulus values of the n vectors are the same, the n vectors can be equally divided on a circle, as shown in Figure 1.

In Figure 1, $F_{1},F_{2},\ldots ,F_{n}$ are n vectors with the same angle. $F_{i}$ represents the i-th complex component of the non-main diagonal elements in the determinant, subject to the following constraints:(7)$$\overset{n}{\underset{i=1}{\sum }}F_{i}=0$$

When the phases of the n complex numbers satisfy that their vector angles are all $\left(\frac{2\pi }{n}\right)$, the sum of the non-main diagonal elements is equal to zero.

In addition, the problem of acquiring the relationship between the phases of the channel matrix and the channel capacity becomes a problem of solving an inhomogeneous linear equation system, as shown in Equation (8):(8)Aφ=b
where $\varphi ={\left(\varphi_{11},\varphi_{12},\cdots ,\varphi_{1n},\varphi_{21},\varphi_{22},\cdots ,\varphi_{n1},\varphi_{n2},\cdots ,\varphi_{nn}\right)}^{T},$ and the coefficient matrix A is defined as:(9)$$A={\left(\begin{array}{cccccc} I_{n} & -I_{n} & 0 & \cdots & 0 & 0\\ 0 & I_{n} & -I_{n} & \cdots & 0 & 0\\ \vdots & \vdots & \ddots & \ddots & \vdots & \vdots \\ 0 & 0 & \cdots & I_{n} & -I_{n} & 0\\ 0 & 0 & \cdots & 0 & I_{n} & -I_{n}\\ 0 & 0 & \cdots & 0 & 0 & 0\\ \vdots & \vdots & \cdots & \vdots & \vdots & \vdots \\ 0 & 0 & \cdots & 0 & 0 & 0 \end{array}\right)}_{nC_{n}^{2}\times n^{2}}$$
where $I_{n}$ represents the *n*-order unit matrix, the size of matrix A is $nC_{n}^{2}\times n^{2}$, the rank of the coefficient matrix is n(n−1). The constant column vector b can be expressed as:(10)$$b={\left(\underset{n-1}{\underbrace{b_{0},b_{0},\ldots ,b_{0}}},\underset{\left(n-1\right)\left(\frac{n}{2}-1\right)}{\underbrace{b_{1},b_{2},\ldots ,b_{c}}},\underset{n^{2}-C_{n}^{2}}{\underbrace{0,0,\ldots ,0}}\right)}^{T}$$
where $b_{0}$ and $b_{l}$ are defined as:(11)$$\{\begin{array}{c} b_{0}={\left(0,\frac{2\pi }{n},\frac{2\cdot 2\pi }{n},\cdots ,\frac{\left(n-1\right)\cdot 2\pi }{n}\right)}^{T}\\ b_{l}=K_{l}\cdot b_{0},l=1,2,\cdots ,c \end{array}$$
where $K_{l}$ is an integer to satisfy linear equation system.

The general solution of the inhomogeneous system of linear equation is equal to the sum of the special solution of linear equations in the inhomogeneous system and the general solution of the homogeneous equation. The special solution $\varphi^{*}$ of this inhomogeneous equation can be written as:(12)$$\varphi^{*}={\left(z_{0},-1\cdot b_{0},-2\cdot b_{0},\ldots ,-\left(n-1\right)\cdot b_{0}\right)}^{T}$$
where $b_{0}$ and $z_{0}$ are defined as:(13)$$\{\begin{array}{c} b_{0}={\left(0,\frac{2\pi }{n},\frac{2\cdot 2\pi }{n},\cdots ,\frac{\left(n-1\right)\cdot 2\pi }{n}\right)}^{T}\\ z_{0}={\left(\underset{n}{\underbrace{0,0,\ldots ,0}}\right)}^{T} \end{array}$$

After finding the special solution, the general solution of the homogeneous linear equation can be written as:(14)Aφ=0

Then the system of equations of the original equation with the same solution can be written as:(15)$$\{\begin{array}{c} \varphi_{11}-\varphi_{21}=0\\ \varphi_{12}-\varphi_{22}=0\\ \vdots \\ \varphi_{1n}-\varphi_{2n}=0\\ \vdots \\ \varphi_{n-1,n}-\varphi_{nn}=0 \end{array}$$

The degree of freedom is $\gamma =n^{2}-n\left(n-1\right)=n$. So it can be written as n linear-independent basis vectors:(16)$$\left(\begin{array}{c} \varphi_{n1}\\ \varphi_{\begin{array}{c} n2\\ \vdots \end{array}}\\ \varphi_{nn} \end{array}\right)=\left(\begin{array}{c} 1\\ \begin{array}{c} 0\\ \vdots \end{array}\\ 0 \end{array}\right),\left(\begin{array}{c} 0\\ \begin{array}{c} 1\\ \vdots \end{array}\\ 0 \end{array}\right),\cdots ,\left(\begin{array}{c} \begin{array}{c} 0\\ 0 \end{array}\\ \vdots \\ 1 \end{array}\right)$$

Then the basic solution of the original equation can be written as:(17)$$\{\begin{array}{c} y_{1}={\left(\underset{n}{\underbrace{1,0,\cdots ,0}},\underset{n}{\underbrace{1,0,\cdots ,0}},\cdots ,\underset{n}{\underbrace{1,0,\cdots ,0}}\right)}^{T}\\ y_{2}={\left(\underset{n}{\underbrace{0,1,\cdots ,0}},\underset{n}{\underbrace{0,1,\cdots ,0}},\cdots ,\underset{n}{\underbrace{0,1,\cdots ,0}}\right)}^{T}\\ \vdots \\ y_{n}={\left(\underset{n}{\underbrace{0,0,\cdots ,1}},\underset{n}{\underbrace{0,0,\cdots ,1}},\cdots ,\underset{n}{\underbrace{0,0,\cdots ,1}}\right)}^{T} \end{array}$$
where $y_{i}$ is the i-th column vector with the size of $n^{2}\times 1$. The general solution of this inhomogeneous equation is given as:(18)$$\varphi =\varphi^{*}+\overset{n}{\underset{i=1}{\sum }}k_{i}y_{i}$$
where $k_{i}$ is an arbitrary constant.

## 3. Analysis of Antenna Configuration

In short-distance transmission, signals are transmitted in LoS. The n×n MIMO system is shown in Figure 2.

In Figure 2, the red graph indicates the receiving antenna, the green one indicates the transmitting antenna. $R_{i}$ is the i-th receiving antenna, $T_{j}$ is the j-th transmitting antenna, and $d_{ij}$ is the sub-channel propagation path.

The phase in the channel matrix H is related to the transmission distance and signal wavelength. When the channel matrix is determined by a LoS component and does not vary with time, the phases of the channel can be expressed as:(19)$$\varphi_{ij}=2\pi \frac{d_{ij}}{\lambda },1\leq i,j\leq n$$
where $d_{ij}$ is the transmission distance and λ is the wavelength.

### 3.1. Circular Arc 2 × 2 Transceiver Antenna Array

The channel phase of a MIMO system needs to satisfy Equation (18) to reach maximum channel capacity. Moreover, in a 2 × 2 antenna array system, the phases can be estimated as:(20)$$\varphi ={\left(0,0,0,-\pi \right)}^{T}+k_{1}{\left(1,0,1,0\right)}^{T}+k_{2}{\left(0,1,0,1\right)}^{T}$$
where $k_{1},k_{2}$ are arbitrary constants. The channel matrix can be written as:(21)$$H=\left(\begin{array}{cc} e^{jk_{1}} & e^{jk_{2}}\\ e^{jk_{1}} & e^{j\left(k_{2}-\pi \right)} \end{array}\right)$$

Then, the maximum channel capacity can be estimated as:(22)$$C_{max}=2log_{2}\left(1+\rho \right)$$
where ρ is SNR.

A 2 × 2 transceiver antenna array can be designed as a circular arc antenna array according to Equation (21), as shown in Figure 3.

According to Figure 3, $d_{s}$ is the array element spacing, the receiving antenna is a circular arc array with a radius of $d_{11}$, and the transmitting antenna is a linear array. d is the distance of the transceiver antennas. $d_{11},d_{12},d_{21},\;\mathrm{and}\;d_{22}$ are sub-channel transmission paths.

Let us assume that:(23)$$\{\begin{array}{c} d_{s}=n\lambda \\ d=md_{s}=mn\lambda \end{array},m>0,n>0$$

Then the sub-channel transmission distance in the circular arc antenna array can be calculated as:(24)$$\{\begin{array}{c} d_{11}=d_{21}=n\lambda \sqrt{1+m^{2}}\\ d_{12}=n\lambda \sqrt{2+m^{2}}\\ d_{22}=mn\lambda \end{array}$$

When m≫1, $d_{11}=d_{21}\approx d_{12}\approx d_{12}$. The phase difference changes dramatically due to the small distance difference Δd, which cannot be ignored. Then the channel matrix can be rewritten as:(25)$$H=\left(\begin{array}{cc} e^{j\varphi_{11}} & e^{j\varphi_{12}}\\ e^{j\varphi_{21}} & e^{j\varphi_{22}} \end{array}\right)$$

According to Equations (21) and (25), if phases $\varphi_{ij}$ of matrix H satisfied Equation (26), the channel capacity reaches its maximum value.
(26)$$\{\begin{array}{c} \varphi_{12}-\varphi_{22}=\pi \\ \varphi_{11}=\varphi_{21} \end{array}$$

According to Equations (19), (24), and (26), Δd can be derived as:(27)$$\Delta d=n\lambda \sqrt{2+m^{2}}-mn\lambda =\frac{1}{2}\lambda +k\lambda ,k\in N$$

Therefore, the expression for m is obtained as:(28)$$m=\frac{2n}{1+2k}-\frac{1+2k}{4n},k\in N$$

According to Equation (28), it is clear that m reaches the maximum value when k=0.
(29)$$m_{max}=2n-\frac{1}{4n}$$

Therefore, according to Equations (23) and (29), it can be derived as:(30)$$d=F\left(\lambda ,d_{s}\right)=\frac{2d_{s}^{2}}{\lambda }-\frac{\lambda }{4}$$

To sum up, Equation (30) is a general expression for 2 × 2 LoS MIMO systems in the circular arc array.

### 3.2. Rectangular 2 × 2 Transceiver Antenna Array

The rectangular array is simpler to configure than the circular arc array in wireless communication systems. The rectangular transceiver antenna array is shown in Figure 4.

In Figure 4, $d_{s}$ is the array element spacing, d shows the distance of the transceiver antennas, and $d_{11},\;d_{12},\;d_{21},\;\mathrm{and}\;d_{22}$ are the sub-channel propagation paths.

According to Equations (4), (19), and (23), the expression of channel capacity can be simplified as:(31)$$C=log_{2}\left\{{\left(\rho +1\right)}^{2}-\frac{\rho^{2}}{2}\left[1+cos\left(4\pi n\left(\sqrt{1+m^{2}}-m\right)\right)\right]\right\}$$

If it satisfies Equation (32), the channel capacity of the rectangular array can also reach the maximum value.
(32)$$cos\left(4\pi n\left(\sqrt{1+m^{2}}-m\right)\right)=-1$$

Then, the relationship between m and n can be represented by Equation (33):(33)$$m=\frac{2n}{2k+1}-\frac{2k+1}{8n},k\in N$$

According to Equation (33), it is clear that m reaches the maximum value when k=0.
(34)$$m_{max}=2n-\frac{1}{8n}$$

According to Equations (23) and (34), the expressions can be simplified as:(35)$$d=F\left(\lambda ,d_{s}\right)=\frac{2d_{s}^{2}}{\lambda }-\frac{\lambda }{8}$$

In summary, Equation (35) is a general expression for 2 × 2 LoS MIMO systems in the rectangular array, which corresponds to the parameters setting in [43].

### 3.3. Circular Arc 3 × 3 Transceiver Antenna Array

A 3 × 3 circular arc transceiver antenna array has a similar calculation method to the 2 × 2 ones. According to Equation (18), we can obtain one of the optimal channel matrices as:(36)$$H_{3\times 3}(\begin{array}{ccc} 1 & 1 & 1\\ 1 & e^{-j\frac{2\pi }{3}} & e^{-j\frac{4\pi }{3}}\\ 1 & e^{-j\frac{4\pi }{3}} & e^{-j\frac{8\pi }{3}} \end{array})$$

Then, the maximum channel capacity can be estimated as:(37)$$C_{max}=3log_{2}\left(1+\rho \right)$$

Therefore, the structure of the antenna is designed in Figure 5.

In Figure 5, $T_{1}$, $T_{2}$, and $T_{3}$ are transmitting antennas. $R_{1}$, $R_{2}$, and $R_{3}$ are receiving antennas. The antennas $T_{2}$, $T_{3}$, $R_{2}$, and $R_{3}$ form a parallelogram. θ is the central angle of the circle with radius r of $T_{1}$, where r is the distance between the transmitting and receiving antennas.

This structure satisfies Equations (38) and (39):(38)$$R_{1}T_{1}=R_{1}T_{2}=R_{1}T_{3}=R_{2}T_{1}=R_{3}T_{1}=r$$
(39)$$\{\begin{array}{c} R_{2}T_{2}=r+\Delta d_{0}\\ R_{2}T_{3}=R_{3}T_{2}=r+\Delta d_{1}\\ R_{3}T_{3}=r+\Delta d_{2} \end{array}$$

According to Equation (19), it can be calculated as:(40)$$\{\begin{array}{c} \Delta d_{0}=-\frac{1}{3}\lambda +k\lambda ,k\epsilon Z\\ \Delta d_{1}=-\frac{2}{3}\lambda +k\lambda ,k\epsilon Z\\ \Delta d_{2}=-\frac{4}{3}\lambda +k\lambda ,k\epsilon Z \end{array}$$

According to this antenna structure, it is necessary to meet the following formula:(41)$$\Delta d_{0}<\Delta d_{1}<\Delta d_{2}$$

Therefore, it can be obtained as:(42)$$\{\begin{array}{c} R_{2}T_{2}=r\sqrt{5-4cos\theta }=r+\frac{2}{3}\lambda +k_{0}\lambda ,k_{0}\epsilon Z\\ R_{2}T_{3}=R_{3}T_{2}=r\sqrt{3-2cos2\theta }=r+\frac{4}{3}\lambda +k_{1},k_{1}\epsilon Z\\ R_{3}T_{3}=r\sqrt{5-4cos2\theta }=r+\frac{5}{3}\lambda +k_{2}\lambda ,k_{2}\epsilon Z \end{array}$$
where $0\leq k_{0}\leq k_{1}\leq k_{2}$. In this configuration, the array element spacing and the distance of the transceiver antennas can be considered as rtanθ and r. Since there is no exact solution to the above equations, an approximate solution can be obtained by allowing a 5% error. Since there exists multiple sets of solutions, a set of solutions with the smallest θ is picked to obtain a smaller antenna scale. The specific results are shown in Table 1.

Table 1 shows that when the distance r is relatively large, there exists an optimal solution for a 3 × 3 transceiver antenna array. The central angle θ is related to the signal frequency. It can be noticed that the array element spacing decreases as the signal frequency increases. Moreover, the comparable solution for the n×n antenna array may be solved in a similar manner, and the ideal antenna configuration can be predicted.

### 3.4. Rectangular 3 × 3 Transceiver Antenna Array

A 3 × 3 rectangular transceiver antenna array has a similar calculation method to the 2 × 2 ones. The rectangular transceiver antenna array is shown in Figure 6.

In Figure 6, $d_{s}$ is the array element spacing, d shows the distance of the transceiver antennas, and $d_{11},\;d_{12},\;d_{21},\;\mathrm{and}\;d_{22}$ are the sub-channel propagation paths.

According to the knowledge of geometry, the transmission distance can be written as:(43)$$\{\begin{array}{c} d_{11}=d_{22}=d_{33}=d\\ d_{21}=d_{12}=d_{32}=d_{23}=\sqrt{d_{s}^{2}+d^{2}}\\ d_{13}=d_{31=}\sqrt{4d_{s}^{2}+d^{2}} \end{array}\;$$

According to Equations (19), (23), and (43), the expression of channel matrix can be written as:(44)$$H_{3\times 3}=\left(\begin{array}{ccc} e^{j2\pi mn} & e^{j2\pi n\sqrt{1+m^{2}}} & e^{j2\pi n\sqrt{4+m^{2}}}\\ e^{j2\pi n\sqrt{1+m^{2}}} & e^{j2\pi mn} & e^{j2\pi n\sqrt{1+m^{2}}}\\ e^{j2\pi n\sqrt{4+m^{2}}} & e^{j2\pi n\sqrt{1+m^{2}}} & e^{j2\pi mn} \end{array}\right)$$

Equation (44) is a very complicated matrix, and there is no precise expression for the channel capacity of the system. However, the channel capacity depends on m and n, which represent the transceiver distance d and array element spacing $d_{s}$ respectively. Moreover, the channel matrix does not satisfy the phase condition Equation (18), and when m≫n, the channel capacity will be close to the minimum value, which can be written as:(45)$$C_{min}=log_{2}\left(1+3\rho \right)$$

The reason why the channel capacity will be close to the minimum value is that when m is large, the effect of n on the channel phase is negligible, which results in a small phase difference of elements in the channel matrix. However, when m is relatively small, an appropriate value of n can be selected to obtain larger channel capacity.

## 4. Performance of Different Antenna Configurations

According to the antenna configurations, rectangular arrays have similar conclusions to circular arc arrays. For a more in-depth study, we can define the aspect ratio ξ.
(46)$$\xi =m=\frac{d}{d_{s}}$$
where ξ is defined as the ratio of distance d of transceiver antennas and the array element spacing $d_{s}$.

### 4.1. Different Performance of 2 × 2 Antenna Configurations

To compare and verify the difference of the 2 × 2 antenna configuration between the arc array and rectangular array, MATLAB simulations were carried out at different ξ with the same SNR of 10 dB.

According to Figure 7a,b, when ξ≫1, whether it is an arc array or a rectangular array, they can reach the maximum capacity. There are differences between the two configurations when the aspect ratio ξ is small, and they are basically the same when the aspect ratio ξ is large, which means that the performance of the two configurations is similar when the distance between the transmitting and receiving antennas is large. The result of the research on the 2 × 2 MIMO system can be calculated as ξ=2n in [43], which uses an approximation of the Taylor expansion in sub-channel propagation paths. This paper gives a more precise expression $\xi =2n-\frac{1}{8n}$ on the 2 × 2 MIMO system, such errors cannot be ignored in short-distance transmission. However, the aspect ratio ξ is approximately equal to 2n in long-distance transmission, where n is the ratio of array element spacing $d_{s}$ to wavelength λ.

According to Equations (29) and (34), Table 2 gives the best value of the aspect ratio ξ and array element spacing $d_{s}$ at the maximum channel capacity. Figure 8 shows the specific change trend.

In Figure 8, two antenna configurations have the same trend of change. In Table 2, the last location of aspect ratio ξ at the maximum channel capacity is given, which corresponds to the last peak in Figure 8. After this peak, the channel capacity will continuously decrease until the lower capacity limit. However, there are still other peaks before the last peak in Figure 8, which are caused by the changes in channel phases in short-distance communications. In addition, the number of peaks occurs more frequently in the rectangular configuration, which means that the channel capacity will change greatly when the distance between the transmitting and receiving antennas changes slightly. Similar results are also given in [40], but this paper gives the location of the last peak, thereby designing the optimal antenna array.

### 4.2. Different Performance of 3 × 3 Antenna Configurations

Unlike the 2 × 2 antenna configuration, the 3 × 3 antenna array configuration has a more complex channel matrix, which results in an approximate solution. To compare the difference of the 3 × 3 antenna configuration between the arc array and rectangular array, MATLAB simulations were carried out at different ξ with the same SNR of 10 dB. Figure 9 shows the simulation results at different aspect ratios.

Figure 9 shows the trend of channel capacity for different 3 × 3 antenna array configurations. Both configurations will produce peaks at different aspect ratios, and the peaks will move backward as the distance of the antenna elements increases. For the circular arc antenna array, it needs to meet Equation (42), which cannot be satisfied in some cases, as shown by the red line in Figure 9a, which means that the configuration cannot achieve the optimal channel capacity in this case. For rectangular antenna array, there are many peaks in Figure 9b, which means that the channel fluctuates a lot, especially in short distance transmissions. In this case, the rectangular configuration is sensitive to small distance changes within the channel and is not suitable for short-distance transmission. On the contrary, the circular arc array changes relatively gently and it is not sensitive to small distance changes, so it is more suitable for short-distance transmission. Moreover, due to the particularity of the circular arc configuration, the spacing of the antenna array elements can be set to be smaller under the same conditions, especially when the scale of the antenna is large.

## 5. Applications

### 5.1. Applications in Coupling Trains

The existing coupling communication is mainly divided into two types. The communication between carriages is referred to as coupling, while the connectivity between trains is referred to as virtual coupling. The virtual coupling train has a longer communication distance than the coupling train. The above conclusion can be applied to the coupling trains of 2 × 2 and 3 × 3 antenna arrays for wireless communication, as shown in Figure 10 and Figure 11.

In Figure 10, the optimal array element spacing $d_{s}$ and the signal frequency f can be determined at a certain distance d of transceiver antennas according to Equation (35) to obtain the largest channel capacity. Table 3 gives the results of 5G potential frequencies.

In Figure 11, the optimal array element spacing $d_{s}$ and the signal frequency f can be determined at a certain distance d of transceiver antennas according to Equation (42) to obtain the largest channel capacity. Table 4 gives the results of 5G potential frequencies.

Table 3 and Table 4 provide the signal frequency, antenna element spacing, and transceiver antenna distance parameters. According to antenna configurations proposed in this paper, the 2 × 2 antenna array is more suitable for the communication of coupling trains, while the 3 × 3 antenna array is more suitable for virtual coupling trains. In addition, as the signal frequency increases, the antenna array element spacing must decrease, making it more ideal for short-range communication.

### 5.2. Applications in Inter-Satellite Communication

The above theory can be applied not only to short-distance LoS coupling train communication, but also to inter-satellite communication in a non-scattering LoS environment. In inter-satellite communication, the distance of transceiver antennas is relatively farther, which means that the aspect ratio ξ is large. In order to ensure the proper spacing of the antenna elements, the signal transmission frequency needs to be increased. However, the frequency of satellite communication is primarily focused on the Ka frequency range (26–40 GHz) and frequencies will be higher in the future. The 2×2 MIMO inter-satellite communication system is shown in Figure 12.

The proper array element spacing $d_{s}$ and the signal frequency f can be determined at a long distance d of transceiver antennas according to Equation (35) to obtain the largest channel capacity. Table 5 provides recommendations for antenna configurations in inter-satellite communication.

As shown in Table 5, when the signal wavelength λ is constant, the larger the array element spacing $d_{s}$ is, the longer the transmission distance d will be. In other words, we can choose an appropriate signal wavelength λ according to the propagation distance d and the array element spacing $d_{s}$. In inter-satellite communication, due to the long distance between the transmitting and receiving antennas, it is necessary to appropriately increase the distance between the antenna elements and the signal frequency to obtain the maximum channel capacity.

## 6. Conclusions

In this paper, the element phase condition is determined when the MIMO system’s n-order channel capacity reaches its maximum value. Moreover, the optimal placement of the 2 × 2 and 3 × 3 transceiver antennas is proposed in the LoS non-scattering environment, which provides basic understanding of n×n optimal antenna placement. The impact of different model parameters on the channel capacity is investigated, including array element spacing, wavelength, and the distance between transmitted and receiving antenna arrays. This theory applies to a channel without scattering and multipath, which is mainly used in short-range LoS communication and a non-scattering environment communication, such as coupling train communication and inter-satellite communication. The simulation results show that we can choose an appropriate signal wavelength and the array element spacing at a certain propagation distance. Moreover, the 2 × 2 antenna array is more suitable for the communication of coupling trains, while the 3 × 3 antenna array is more suitable for virtual coupling trains according to proposed antenna configurations. Furthermore, the 2 × 2 antenna rectangular configuration proposed in this paper has reached the optimal channel in inter-satellite communication, which lays a foundation for the deployment of communication systems. However, according to the theory proposed in this paper, the optimal channel can still be achieved in larger scale MIMO systems, but it needs to correspond to different antenna configurations, which needs to be studied further.

## Figures and Tables

**Figure 1 sensors-22-03669-f001:**
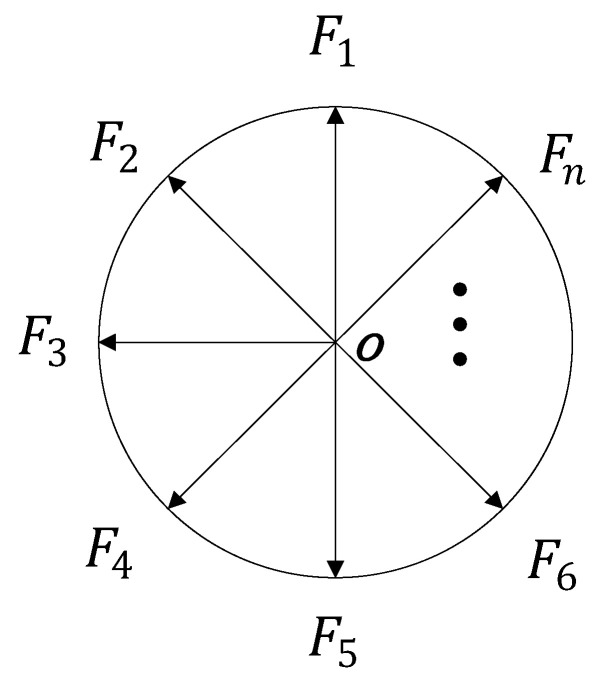
Schematic of n vectors with the same angle.

**Figure 2 sensors-22-03669-f002:**
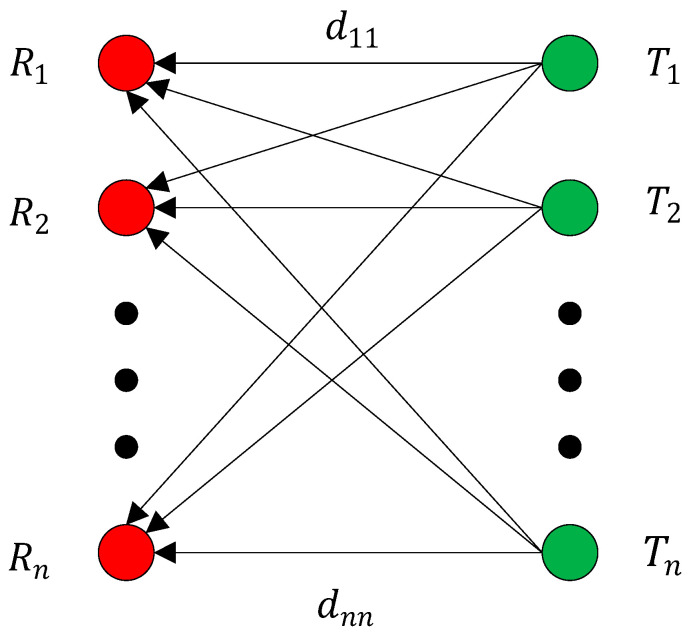
Schematic representation of n×n channel.

**Figure 3 sensors-22-03669-f003:**
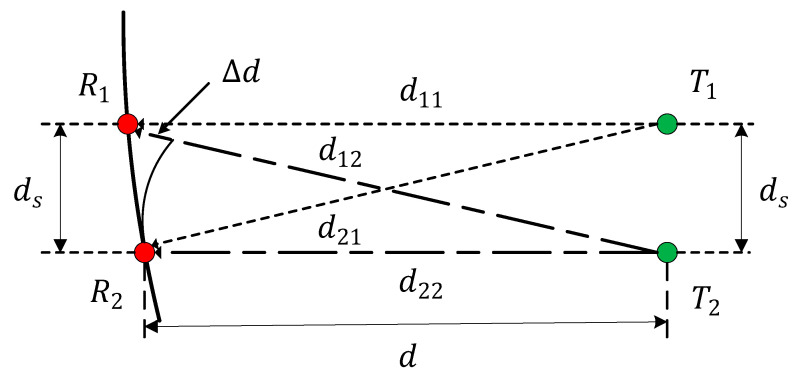
Circular arc 2 × 2 transceiver antenna array.

**Figure 4 sensors-22-03669-f004:**
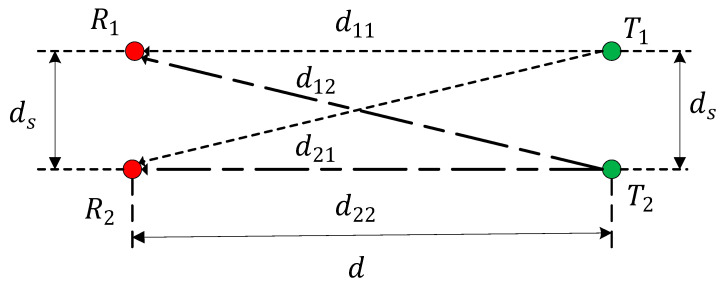
Rectangular 2 × 2 transceiver antenna array.

**Figure 5 sensors-22-03669-f005:**
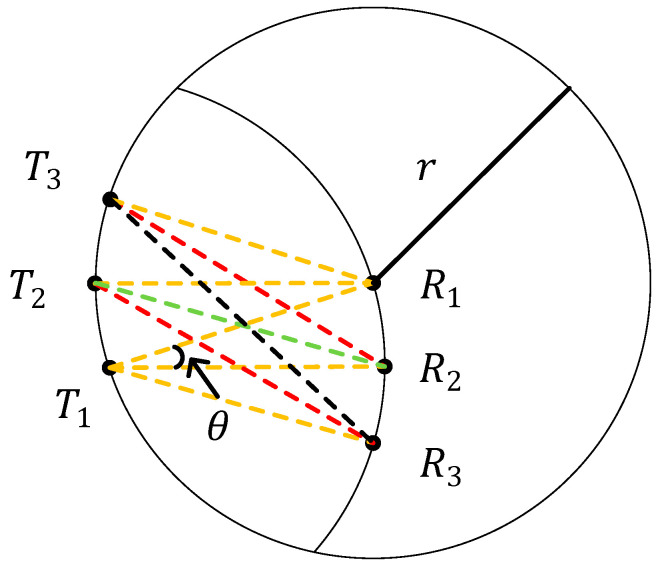
Circular arc 3 × 3 transceiver antenna array.

**Figure 6 sensors-22-03669-f006:**
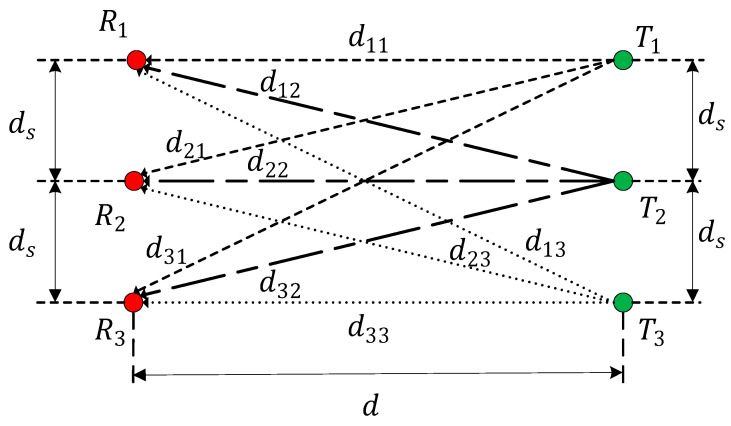
Rectangular 3 × 3 transceiver antenna array.

**Figure 7 sensors-22-03669-f007:**
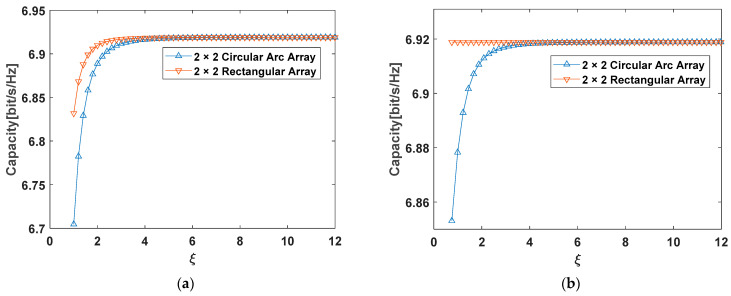
Channel capacity at different aspect ratios ξ:$\;\left(a\right)\;\xi =2n;\left(b\right)\;\xi =2n-\frac{1}{8n}$.

**Figure 8 sensors-22-03669-f008:**
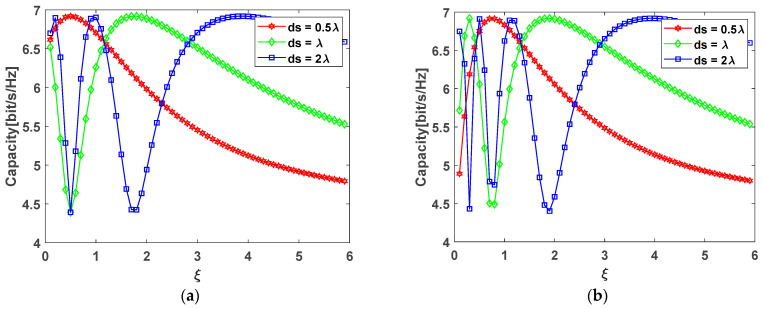
Channel capacity under two different 2 × 2 antenna configurations: (**a**) circular arc antenna array; (**b**) rectangular antenna array.

**Figure 9 sensors-22-03669-f009:**
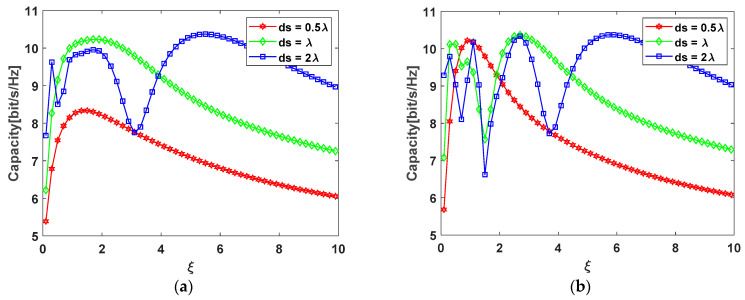
Channel capacity under two different 3 × 3 antenna configurations: (**a**) circular arc antenna array; (**b**) rectangular antenna array.

**Figure 10 sensors-22-03669-f010:**
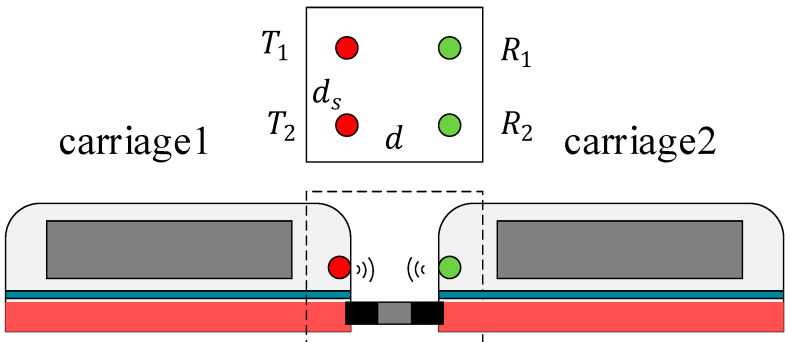
The coupling train of 2×2 MIMO system.

**Figure 11 sensors-22-03669-f011:**
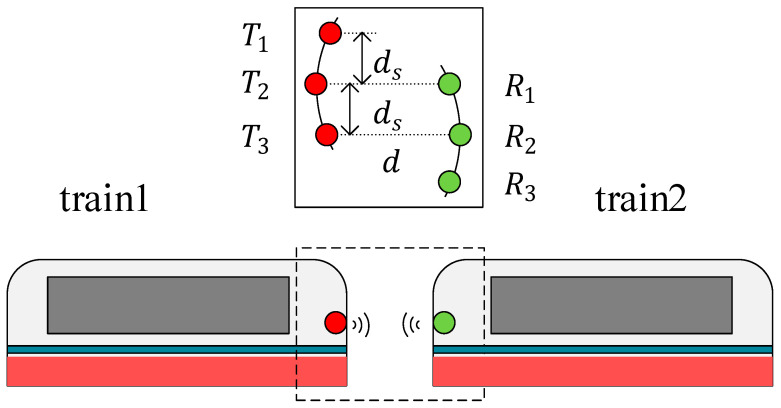
The virtual coupling train of the 3×3 MIMO system.

**Figure 12 sensors-22-03669-f012:**
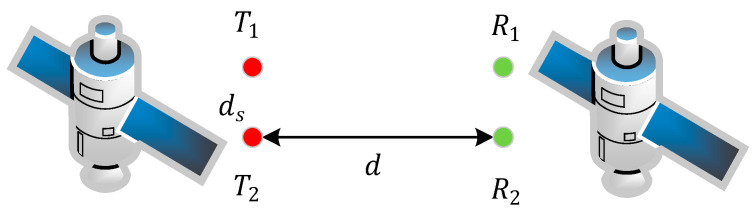
The 2×2 MIMO inter-satellite communication system.

**Table 1 sensors-22-03669-t001:** Solution of the system equations.

Signal Frequency f (GHz)	Distance r	$k_{0}$	$k_{1}$	$k_{2}$	θ (rad)
2.6	10	0	0	1	0.0879
	100	0	0	1	0.0277
	300	0	0	1	0.0160
3.5	10	0	0	1	0.0757
	100	0	0	1	0.0239
	300	0	0	1	0.0138
5.6	10	0	0	1	0.0598
	100	0	0	1	0.0189
	300	0	0	1	0.0109
28	10	0	0	1	0.0267
	100	0	0	1	0.0085
	300	0	0	1	0.0049

**Table 2 sensors-22-03669-t002:** Parameters of theoretical maximum capacity.

$d_{s}\;\left(\lambda \right)$	ξ	Antenna Configuration
0.5	0.5	Circular Arc
1	1.75	Circular Arc
2	3.875	Circular Arc
0.5	0.75	Rectangular
1	1.875	Rectangular
2	3.9375	Rectangular

**Table 3 sensors-22-03669-t003:** Recommendations for 2 × 2 MIMO system of the coupling train.

f (GHz)	λ (m)	d (m)	$d_{s}\;\left(m\right)$
2.6	0.1154	0.5	0.1723
2.6	0.1154	1	0.2419
3.5	0.0857	0.5	0.1479
3.5	0.0857	1	0.2081
5.6	0.0536	0.5	0.1165
5.6	0.0536	1	0.1643
28	0.0107	0.5	0.0518
28	0.0107	1	0.0732

**Table 4 sensors-22-03669-t004:** Recommendations for 3 × 3 MIMO system of virtual coupling train.

f (GHz)	λ (m)	d (m)	$d_{s}\;\left(m\right)$
2.6	0.1154	10	0.8779
2.6	0.1154	20	1.2409
3.5	0.0857	10	0.7565
3.5	0.0857	20	1.0694
5.6	0.0536	10	0.5979
5.6	0.0536	20	0.8453
28	0.0107	10	0.2673
28	0.0107	20	0.3780

**Table 5 sensors-22-03669-t005:** Recommendations in inter-satellite communication.

f (GHz)	λ (m)	d (m)	$d_{s}\;\left(m\right)$	ξ
30	0.01	50	0.05	100
30	0.01	5000	0.5	1000
40	0.0075	67	0.05	133
40	0.0075	6667	0.5	1333
75	0.004	125	0.05	250
75	0.004	12,500	0.5	2500
300	0.001	500	0.05	1000
300	0.001	50,000	0.5	10,000

## Data Availability

Not applicable.

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
