# Peer review of "Design and Analysis of High-Capacity MIMO System in Line-of-Sight Communication"

_sensors, 2022, doi:10.3390/s22103669_

Round 1

Reviewer 1 Report

The authors determine the general expression for the phase of the channel matrix at maximum channel capacity. Moreover, the optimal antenna configuration of 2×2 and 3×3 transceiver antenna array is realized for LoS communication, providing methods for ? × ? optimal antenna placement. Simulation results show that the 2×2 antenna array is more suitable for the
communication of coupling trains, while the 3×3 antenna array is more suitable for virtual coupling trains according to antenna configurations proposed in this paper, which provides basic knowledge for the deployment of coupling trains communication systems. 

This work is written and organized well. I suggest some revisions that can further improve this paper:

1) The Introduction section should be split into subsections, i.e., Related Literature, Motivation, Main Contribution, etc.

2) If the authors make subsections in Section 1, it is better to also summarize the main contributions of this work in subsection "Motivation and Contributions".

3) This paper does not discuss any system model, various assumptions, and considerations. What the system model consists of? what are the assumptions? and considerations?

4) The text in all figures is very big. It should be aligned with the paper text size. 

5) The results discussion is too short, a more detailed discussion is required.

6) It is suggested to add one more paragraph in the Introduction section (should be the first paragraph) about emerging 5G/6G features, technologies, and what will be the challenges for 5G/6G.

7) Some promising next-generation technologies are IRS, backscatter communications, UAV, NOMA, and so on. These should be discussed and referred such as Opportunities for physical layer security in UAV communication enhanced with intelligent reflective surfaces; Energy-efficient IRS-aided NOMA beamforming for 6G wireless communications; NOMA-enabled backscatter communications for green transportation in automotive-industry 5.0; etc.

8) This paper reports very old papers in the literature, recent literature (i.e., 2020,2021,2022) should be studied and discussed in this paper. Then based on these latest works, the motivation and main contributions of this work should be revised in detail.

9) There are also found some grammar errors and typos in this manuscript. The authors need extensive proofreading of this paper before submitting the revised version of this paper.

Reviewer 2 Report

Overall, the proposed results and analysis has been described quite well in the manuscript. There are quite a few typographical errors in the manuscript that should be relatively easy to correct, such as the "T3" label in figure 2 (I assume should instead be "Tn"), along with numerous pairs of words that are joined together when there should be a space between them.

In Section 3, the results were described for any general nxn MIMO system and a detailed analysis was conducted for the case of a 2x2 system, both for circular arc and a rectangular transceiver array. However, the results were only extended for the case of the 3x3 circular arc transceiver array. It is not clear why the same extension was not included for the case of a 3x3 rectangular array. Furthermore, the performance analysis in Section 4 was limited to the case of a 2x2 MIMO array. It would be worthwhile to extend the performance analysis in Section 4 to highlight the differences between the 2x2 and 3x3 systems for both the circular arc and rectangular transceiver arrays. This would allow the reader to observe any possible trends between the 2x2 and 3x3 systems.

Round 2

Reviewer 1 Report

Thank you so much for addressing all of my comments. This article is improved now and I recommend it for publication.

Author Response

Thanks very sincerely for your comments!